# In Vitro Selection and Characterization of HIV-1 Variants with Increased Resistance to LP-40, Enfuvirtide-Based Lipopeptide Inhibitor

**DOI:** 10.3390/ijms23126638

**Published:** 2022-06-14

**Authors:** Yue Hu, Wenjiang Yu, Xiuzhu Geng, Yuanmei Zhu, Huihui Chong, Yuxian He

**Affiliations:** 1NHC Key Laboratory of Systems Biology of Pathogens, Institute of Pathogen Biology, Chinese Academy of Medical Sciences and Peking Union Medical College, Beijing 100730, China; huyue@ipbcams.ac.cn (Y.H.); yuwenjiang@ipbcams.ac.cn (W.Y.); gengxiuzhu@ipbcams.ac.cn (X.G.); zhuym@ipbcams.ac.cn (Y.Z.); 2Center for AIDS Research, Chinese Academy of Medical Sciences and Peking Union Medical College, Beijing 100730, China

**Keywords:** HIV-1, gp41, fusion inhibitor, resistance, LP-40, T20

## Abstract

In our previous work, we replaced the TRM (tryptophan-rich motif) of T20 (Enfuvirtide) with fatty acid (C16) to obtain the novel lipopeptide LP-40, and LP-40 displayed enhanced antiviral activity. In this study, we investigated whether the C16 modification could enhance the high-resistance barrier of the inhibitor LP-40. To address this question, we performed an in vitro simultaneous screening of HIV-1_NL4-3_ resistance to T20 and LP-40. The mechanism of drug resistance for HIV-1 Env was further studied using the expression and processing of the Env glycoprotein, the effect of the Env mutation on the entry and fusion ability of the virus, and an analysis of changes to the gp41 core structure. The results indicate that the LP-40 activity is enhanced and that it has a high resistance barrier. In a detailed analysis of the resistance sites, we found that mutations in L33S conferred a stronger resistance, except for the well-recognized mutations in amino acids 36–45 of gp41 NHR, which reduced the inhibitory activity of the CHR-derived peptides. The compensatory mutation of eight amino acids in the CHR region (NDQEEDYN) plays an important role in drug resistance. LP-40 and T20 have similar resistance mutation sites, and we speculate that the same resistance profile may arise if LP-40 is used in a clinical setting.

## 1. Introduction

AIDS, acquired immunodeficiency syndrome, is a chronic infectious disease that is caused by human immunodeficiency virus infection, which is mainly characterized by damage to the body’s immune system [1,2]. The outer layer of the virion is a lipoprotein envelope in which two virus-specific envelope glycoproteins, gp120 and gp41, are embedded [3,4,5]. When the virus infects the target cells, gp120 binds to the CD4 receptor on the target cell membrane, resulting in a conformational change of gp120. Following this, it binds to the accessory receptors CXCR4 or CCR5 on the target cell membrane, inducing a separation of HIV gp120 from the transmembrane protein gp41 [6,7,8], which then begins to change its conformation and exposes its amino-terminal fusion peptide (FP). FP is then inserted into the target cell membrane, forming a pre-hairpin intermediate structure that bridges the viral and target cell membranes. Subsequently, the three gp41 NHR (N-terminal heptad repeat) sequences interact to form a trimeric hydrophobic coiled core, while the three CHRs (C-terminal heptad repeats) bind in reverse parallel in a highly conserved, hydrophobic amino acid groove formed by the NHR [9,10]. The two interact to form a closely arranged hexamer helix bundle (six-helix bundle, 6-HB), which fuses the virus membrane to the cell membrane, allowing the virus to enter the target cell. Extensive data suggest that the peptides synthesized by NHR or CHR of GP41 can competitively bind the pre-hairpin intermediate of GP41 and prevent the formation of viral 6-HB structures. This is considered to be an ideal target for the development of anti-HIV drugs [11].

T20 (Enfuvirtide/fuzeon), a CHR-derived peptide containing 36 amino acids, was approved for clinical use in 2003, and it was the first type of HIV entry inhibitor [10]. T20 has proven its effectiveness in the combination treatment of HIV-1 infection. However, it requires frequent and high doses (90 mg, twice a day), and resistance is easily acquired [12,13,14]. The TRM of the T20 carboxy-terminal, also known as the MPER (membrane proximal external region), is rich in Tyrannine sequences (WASLWNWF). Therefore, it is highly hydrophobic (Figure 1). Our previous work demonstrated that the TRM region has a function of promoting the binding of T20/N39 to form a stable 6-HB structure, and it has some hydrophobic interactions between the FPPR sequence at the corresponding position of gp41 [15]. The interaction between T20 and the target cell membrane is required for its antiviral activity, which appears to depend mainly on the anchoring effect of the TRM region at its carboxy-terminal group on the target cell membrane. We replaced TRM in the T20 sequence with a fatty acid (palmitic acid, C16) that also had high hydrophobicity [16,17]. That is, the Lys side chain of the T20△TRM carboxyl terminal was modified via fatty acid to obtain the lipopeptide LP-40. In a single infection inhibition experiment with the NL4-3_D36G_ virus, the IC_50_ value of LP-40 antiviral activity was increased by approximately seven-fold compared to T20 [15]. C16 modification was able to enhance the antiviral activity of the LP-40 against broad subtypes of HIV-1, and it showed a certain broad-spectrum antiviral activity.

In this study, we investigated whether the replacement of the TRM region of T20 with a C16 modification can increase the high resistance barrier of the inhibitor LP-40. To address this issue, we conducted an in vitro screening of both T20 and LP-40 for drug-resistant HIV-1 NL4-3, and the virus was gradually passed under progressively increasing inhibitor doses. After nearly one year of 32 generations of in vitro resistance screening passages, the concentration of T20 had increased to 2000 µM; in contrast, LP-40 was much more difficult to induce resistance than T20, with LP-40 approaching only 200 µM due to difficulties in viral passages. These results suggest that replacing the TRM region of T20 with a C16 modification may have significantly increased the resistance barrier. We focused on the molecular pathways of T20- and LP-40-induced resistance, with the aim of gaining new insights into the genetic pathways and mechanisms of HIV-1 resistance to newly developed fusion inhibitors. The mechanism of drug resistance was further investigated from various aspects, such as mutant viral Env glycoprotein expression and processing, the effect of the Env mutation on viral entry ability and fusion ability, and the analysis of changes in the core structure of gp41. These data provide key information on the structure–function relationship of HIV-1 Env- and gp41-dependent fusion inhibitors and assist with providing a guiding direction for the development of new anti-HIV drugs.

## 2. Results

### 2.1. Selection of LP-40-Resistant HIV-1 In Vitro

To determine the resistance profile of LP-40, LP-40-resistant variants were selected using a dose-escalating method. The selection for drug-resistant HIV-1NL4-3 commenced with a concentration of 0.04 µM LP-40 (Figure 2B). Finally, after 32 passages, the induced concentration of LP-40 reached 200 µM. As shown in Figure 3, in generation 9 (P-9), the concentration of LP-40 was 0.3 µM, and it was observed that, at position 37 of the gp41 NHR coding region (I37T), isoleucine was replaced by threonine. As shown in passage 9 (P-9), the concentration of LP-40 was 0.3 µM, and it can be observed that isoleucine was replaced by threonine at position 37 in the gp41 NHR coding region (I37T). G86D and C87Y were observed; at position 119 (E 119 G) in the gp41 CHR coding region, glutamate was replaced by glycine; at position 126 (N 126 K), asparagine was replaced by lysine. In P-24, the concentration of LP-40 was 8 µM; I37T in the NHR coding region was still present, and A15V, A21T, and L57H were also present, and N126K at position 126 in the CHR coding region was still present. In P-26, the concentration of LP-40 was 20 µM; at L33S, position 33 in the NHR coding region, leucine was replaced by serine and I37T remained. Additionally, a combination of mutations—E109N, N113D, E119Q, D121E, N125E, S129D, H132Y, and S133N (abbreviated as NDQEEDYN)—appeared in the coding region of CHR. At a concentration of 200 µM, in addition to L33S and I37T, D36G, V38A, V38T, N42T, and I69V appeared in the NHR coding region, and a combination of the above eight mutations still existed in the CHR coding region.

### 2.2. Selection of T20-Resistant HIV-1 In Vitro

To determine the drug resistance profile of T20, we performed a parallel in vitro HIV-1 mutant screen using a dose-escalation approach (Figure 2). Selection for drug-resistant HIV-1_NL4-3_ commenced with a concentration of 0.5 µM T20 (Figure 2A). The selection of viral mutants of T20 was relatively easy to obtain, and the virus was still detectable after the T20 drug concentration was multiplied. However, after 32 passages, the induced concentration was stopped at 2000 µM due to T20 showing solubility problems. The Env gene was amplified from the virus of the culture, and the complete part of gp41 was sequenced. A considerable number of amino acid changes were found in gp41. As shown in Figure 3, in passage 9 (P-9), when T20 is at a concentration of 12 µM, it was observed that, at the 50th position in the gp41 NHR coding region (A50V), alanine was replaced by a valine. At position 71, A71V, alanine was replaced by a valine; at E 137 K, glutamate was replaced by a lysine; and, at position 140, N140S, asparagine was replaced by a serine. In P-21, when the concentration of T20 was 200 µM, the mutated amino acids appearing in the P-9 generation were restored to the original amino acids, and a new set of mutations—L33S, D36G, V38A, R68G, and I69V, and a combination of eight mutation sites (NDQEEDYN) in the CHR coding region appeared in the NHR coding region. In P-24, when the concentration of T20 was 400 µM, the mutations L33S and N42D were observed in the NHR coding region, and N113S and N126K were observed in the CHR coding regions. In P-32, when the concentration of T20 was 2000 µM, a new set of mutant combinations, G20V, S35P, D36G, V38A, N42T N43S, and I69V, appeared in the NHR, and NDQEEDYN in the CHR. These eight mutant sites (NDQEEDYN) have been observed in passages P21 and P32. 

### 2.3. Construction of a Series of Mutant Pseudoviruses to Simulate Viruses under Drug Pressure, and Analysis of the Effect of a Series of Mutations on the Function of Viral Envelope Proteins

To simulate the sequence changes of viral resistance genes under different drug concentration pressures, we constructed the Env genes under different drug concentration pressures in the PCDNA3.1 vector and obtained a series of mutations. T20-V 1~5 was constructed at a concentration of T20 12 µM (P9). T20-V 6~12, T20-V 13~14, and T 20-V 15~18 were constructed at concentrations of 200 µM (P21), 400 µM (P24), and 2000 µM (P32), respectively. Similarly, the LP-40 pressure concentrations were 0.3 µM, 8 µM, 20 µM, and 200 µM, respectively, and the constructs were numbered LP40-V 1~5, LP40-V 6~8, LP40-V 9~11, and LP40-V 12~15. Resistance mutations will cause varying levels of damage [12,18,19] to the function of the pseudoviral envelope proteins. Here, we explored how serial mutations affect the function of viral envelope proteins. Based on this problem, we performed a single round of quantitative P24-infection experiments, packaging the pseudovirus with a previously constructed series of mutant plasmids. We compared the invasion ability of the HIV-1 wild-type and mutant pseudoviruses toward target cells under the same P24 content (using the invasion capacity of each subtype as 100%). 

As shown in Figure 4A, the serial mutations had different invasion abilities. Among the 18 series of mutations with T20-induced resistance, T20-V3 (*p* < 0.05), T20-V8 (*p* < 0.01), and T20-V15 (*p* < 0.01) increased significantly compared to the wild-type strain. In contrast, T20-V5, T20-V9, T20-V11, T20-V12, T20-V17, and T20-V18 showed significant decreases in invasive ability compared to the wild-type strain (statistically different, *p* < 0.0001). T20-V1 (*p* < 0.01), T20-V7 (*p* < 0.05), and T20-V13 (*p* < 0.001) also showed decreases; the others showed no significant differences compared to the wild-type strains. No statistically significant increase in invasive ability was observed in any of the 15 series of LP-40-induced drug resistance mutations (Figure 4B). LP40-V3, LP40-V4, LP40-V5, LP40-V6, LP40-V7, LP40-V11, and LP40-V15 decreased significantly (*p* < 0.0001), and LP40-V10 and LP40-V13 decreased by approximately 50% (*p* < 0.01) compared to the wild-type strain; LP40-V2 and LP40-V9 (*p* < 0.05) decreased only slightly.

Additionally, we compared the differences between the wild-type envelope proteins and the serial mutant proteins in mediating membrane fusion using a double-separated protein-based fusion experimental system. In this assay, 293T cells act as effector cells that express both DSP1-7 and the envelope proteins. The 293T cells stably expressing DSP8-11/CD4/CXCR4/CCR5 were used as target cells. After the fusion of the effector cells and the target cells, the fluorescent reporter could be detected with the assistance of the live cell substrate, and the strength of the fluorescence could also indicate the ability and the rate of fusion. In the analysis of the fusion experimental results, we set the fusion capacity of each subtype WT at 100% to calculate the fusion capacity of both mutant strains. The results are shown in Figure 4C: an almost complete loss of fusion ability in T20-V5, T20-V9, T20-V11, and T20-V12 (*p* < 0.0001) in a series of 18 mutations with T20-induced drug resistance. T20-V1 and T20-V18 (*p* < 0.0001) had a 40% lower fusion ability than the wild-type; T20-V2, T20-V7, T20-V10, and T20-V17 had a 20–40% lower fusion ability than the wild-type; and T20-V6 (*p* < 0.001), T20-V4, T20-V13, T20-V14, T20- V15, and T20-V16 (*p* < 0.05) were all increased compared to the wild strain. Among the 15 series of mutations in which LP-40-induced resistance, LP40-V3, LP40-V5, LP40-V6, LP40-V7, and LP40-V15 (*p* < 0.0001) fusion ability was almost completely lost, and LP40-V4 and LP40-V10 membrane fusion ability was only less than 40% of that of the wild strain. In contrast, LP40-V2, LP40-V8, LP40-V9, LP40-V12, and LP40-V13 had improved fusion ability compared with the wild-type strain (Figure 4D). As shown in Figure 4E,F, we measured the dynamic fusion curves of different mutant strains within 0–6 h to visually and dynamically observe the fusion rates of mutant strains. Among them, we selected the fusion ability of the sixth hour to evaluate the mutant strains. The fusion activity is shown in Figure 4C,D.

HIV has high mutation rates, and quasispecies dynamics are essential features. We obtained Env gene sequences, constructed and obtained pseudoviruses, and found that some of them had lost their invasion or fusion activity, which may be because some of the quasispecies library lost their activity due to mutation.

### 2.4. Expression and Processing of a Series of Mutations in the Env Glycoprotein 

It is crucial to understand whether resistance mutations affect the expression and processing of Env glycoproteins, and whether they affect alterations in viral infectivity and drug resistance. First, we transfected 293T cells with the enveloped protein plasmid to express the gp160 protein. We collected cell precipitates (preparation of the lysate). The Env glycoprotein and Env glycoprotein expression and processing in cells and in the viral supernatant were characterized via Western blotting. Two bands of gp160 and gp120 were detected with an anti-gp120 antibody, two bands of gp160 and gp41 were detected with the 10E8 antibody, and internal reference bands were detected using anti-β-actin or P24. The gp160/gp120/gp41 protein levels in the cell lysate samples can be seen in Figure 5A. T20-V12 caused abnormal protein expression due to base deletion, with uneven levels of gp120 protein in the cell lysates. Among them, the protein expression levels of T20-V4, T20-V6, T20-V10, T20-V14, T20-V15, and T20-V17 were basically the same as that of the WT envelope protein, and all the others were weakened. The protein expression levels of the T20-V8, T20-V9, and T20-V10 mutations were diminished compared to the gp160 protein expression levels of the WT envelope protein. The gp41 protein expression levels of T20-V6, T20-V10, T20-V15, T20-V16, T20-V17, and T20-V10 basically matched the WT envelope protein, but all the others were weakened. Comparing the protein expression of each mutant strain, we found that the N126K mutation in T20-V13 and T20-V15 caused altered glycosylation and caused changes in protein size.

Figure 5B shows the gp160/gp120/gp41 protein levels in the cell lysate samples. No corresponding protein expression was detected via the abnormal protein expression of LP40-V3 and LP40-V5. The mutations in the cell lysates were almost comparable to the gp120/gp160 protein expression levels of the WT envelope protein. The LP40-V1, LP40-V4, LP40-V7, LP40-V8, LP40-V10, LP40-V11, and LP40-V15 mutations showed reduced gp41 protein expression levels compared to the WT envelope protein. We noticed an interesting phenomenon in that a high level of gp41 protein expression corresponded to an enhanced entry or fusion capacity of the viral strains, but reduced protein expression did not necessarily affect entry or fusion capacity.

As shown in Figure 5C, there were gp160/gp120/gp41 proteins in the viral supernatant samples and no gp160 protein expression difference in the viral supernatant, whereas the protein expression level of the WT envelope protein in the T20-V11 and T20-V13 mutations was enhanced compared with gp160. The gp41 protein expression levels of T20-V1, T20-V5, T20-V6, T20-V9, T20-V10, T20-V11, T20-V15, T20-V16, T 20-V17, and T20-V18 mutations diminished.

Figure 5D shows that the viral supernatants had similar levels of gp41 protein expression to Env alone. LP40-V6, LP40-V7, LP40-V9, LP40-V10, LP40-V11, LP40-V14, and LP40-V15 had reduced gp41 protein expression levels compared to WT envelope proteins, consistent with decreased entry and fusion capacity.

### 2.5. Characteristics of Resistance of Serial Mutations to T20 and LP-40

Except for the loss-of-entry pseudoviruses, the corresponding pseudoviruses were produced separately from the series of plasmids to determine the sensitivity and cross-resistance of the series of mutations to T20 and LP-40, and the NL4-3 pseudovirus was also experimented upon simultaneously as a control. As the results show in Table 1, with increasing drug concentrations, the correspondingly screened strains showed strong drug resistance, and mutations in the T20-resistant strains led to a significant decrease in the susceptibility of the pseudovirus to T20, even up to 112-fold. The virus strain induced at T20 12 µM concentrations showed approximately one- to two-fold less sensitivity to T20. In contrast, when the T20 concentration rose to 200 µM, the inhibitory IC_50_ for the viral strains induced by T20 was 2.8–5.3 µM, and the sensitivity decreased by more than 50-fold. Among them, T20 inhibited the T20-V8 strain to 5.3 µM. The frequency of the T20-V8 strain was 1/12. From sequence alignment, except for the L33S, which appeared in all the sequences, the new mutation site of D36G/V38A/N42T was also introduced. When the T20 concentration rose to 2000 µM, the previous L33S disappeared, and D36G/V38A/N42T became the main mutation point, with a 32- to 112-fold lower sensitivity to T20. Among them, N43S appeared in the sequence of the T20-V18 strain, and its IC_50_ by T20 decreased to 6.1 µM.

Mutations in the resistant strain of serial LP-40 resulted in a significant decrease in sensitivity to LP-40, even over 227-fold. The sensitivity of LP-40 was reduced by approximately 1.5–12.6-fold, and the IC_50_ of LP40-V2 reached 0.4 µM at a frequency of 4/9. However, when the concentration of LP-40 rose to 8 µM and 20 µM, the IC_50_ of the viral strains induced by LP-40 was 2.8–3.3 µM, and the sensitivity was reduced more than 92-fold. LP40-V11 strain sequence alignment found that, in addition to I37T, new mutation sites of L33S had also been introduced, and the frequency of the LP40-V11 strain was 3/8. Among them, LP-40-inhibited LP40-V11 reached 3.3 µM. When the concentration of LP-40 rose to 200 µM, I37T disappeared, and L33S/D36G/V38A/N42T/I69V became the main mutation point. The sensitivity to LP-40 decreased by more than 227-fold. The N126K mutation site had a more obvious effect on LP-40 than on T20. Cross-resistance appeared in resistant strains of T20 and LP-40. The resistant strain of T20 caused a 227-fold increase in the resistance of the LP-40, and the resistance of the LP-40 strains resistant to T20 increased 107-fold. When the concentration of T20 was 200 µM, the resistant strain T20-V8 appeared at P21, and the resistance point in the sequence was very similar to that with a 200 µM concentration of LP-40, as shown in P32. This indicates that, in the case of the parallel induction of resistance, T20 showed an earlier alteration of high resistance sites.

### 2.6. Effects of Single-Point Mutations on Viral Entry and Fusion

Twenty-five corresponding single-point mutation plasmids and one eight-point combination (NDQEEDYN) mutation plasmid were constructed to determine the effect of individual point mutations on viral entry via a comparison of the resistance sites in NHR and CHR in gp41 in the T20 and LP-40 drug pressure screens (Figure 6A). The entry abilities of A15V, L57H, R68G, G86D, C87Y, E119G, and N140S were almost completely lost, corresponding to LP40-V6, LP40-V7, T20-V9, LP40-V3, LP40-V5, LP40-V4, and T20-V5, respectively, so it was determined that these mutation sites might be lethal mutations. G20V, A21T, S35P, and Y127H had a more than 50% reduction in entry ability compared with the wild-type strain. D36G, I37T, V38T, I69V, A71V, S129N, E137K, and NDQEEDYN had significantly improved entry capabilities (Figure 6A). The differences between the wild-type envelope proteins and serial mutant proteins in mediating membrane fusion were compared using a double-separated protein-based fusion experimental system. Except for NDQEEDYN and N126K, which had significantly higher fusion ability, and A71V, which had no significant difference, all the other mutations had significantly lower fusion abilities. A15V, L57H, R68G, G86D, and C87Y all showed a complete loss of fusion and entry ability. G20V, A21T, E119G, and Y127H had only an approximately 10% fusion capacity, and L33S, N42D, N42T, and N43S had less than 50% fusion capacity, but N140S had approximately 60% fusion capacity (Figure 6B). As shown in Figure 6C, we measured the dynamic fusion curves of different mutant strains within 0–6 h to visually and dynamically observe the fusion rates of the mutant strains. Among them, we selected the fusion ability of the sixth hour to evaluate the fusion activities of the mutant strains (Figure 6B).

### 2.7. Cross-Resistance to T20 and LP40 of Single-Point Mutations on Pseudoviruses

As shown, the point mutations A15V, L57H, R68G, G86D, C87Y, E119G, and N140S caused significant reductions in the viral entry activity, and viral titers were insufficient for the later evaluation of the drug activity for T20 and LP-40. As shown in Table 2, 18 other point-mutant pseudoviruses were evaluated for T20 and LP-40 sensitivity, respectively. Six point-mutant pseudovirus (L33S, I37T, V38A, V38T, N42T, and N43S) showed resistance to T20, with IC_50_ in the range of 0.41–3.4 µM and a 6–51-fold decrease in sensitivity. Similarly, these six mutant pseudoviruses also showed resistance to LP-40, with IC_50_ in the range of 0.2–3.5 µM and a 6–111-fold decrease in sensitivity. In addition to these six mutation points, two mutation points (G20V and N126K) showed differences between LP-40 and T20. The sensitivity of the G20V mutant pseudovirus to T20 was slightly increased, while the sensitivity to LP-40 IC_50_ reached 0.2 µM, which decreased by 6.5-fold. The mutation G20V only appeared in T20-induced drug resistance, and the frequency was low. The sensitivity of the N126K mutant pseudovirus to T20 was slightly reduced by 1.7-fold, while the sensitivity to LP-40 IC_50_ reached 0.3 µM and decreased 9.7-fold. This mutation point N126K appeared in LP-40-induced drug resistance with a high frequency.

### 2.8. Identification of the Sensitivity of LP-40-Induced Resistance Mutations to Serial Membrane Fusion Inhibitors

We selected LP-40-induced drug resistance typical point mutations and combination mutation pseudoviruses to evaluate the current representative membrane fusion inhibitors. The I37T mutation site was first observed in LP-40-induced resistant P9, and, as shown in Table 3, the membrane fusion inhibitors all showed reduced inhibitory activity against the I37T mutation site, and LP-40 and C34 even reduced the I37T mutation inhibitory activity by more than 17-fold. The I37T/L33S mutation appeared in P26, and the L33S single-point mutation exhibited resistance to most inhibitors, with a stacked multiplicative effect of resistance after the combined mutation. The P32 generation showed L33S/V38A/N42T mutations, and the superimposed effect of drug resistance was even more pronounced, with the sensitivity of C34 even reduced by 1140-fold. Eight combinations (NDQEEDYN) on CHR began to appear in P26, and all the other inhibitors except T20 and T1249 showed a reduced sensitivity to combination mutations. In the combination of the I37T/L33S and NDQEEDYN mutations, all the inhibitors showed significant drug resistance, and their drug resistance to C34 significantly decreased by 846-fold, indicating that the mutant combination could escape the inhibition of C34. Both the L33S/V38A/N42T and NDQEEDYN mutations showed the strongest resistance to the inhibitor LP-40, C34, SFT, T1249, and T2635; T2635 also showed resistance, with a 7.85-fold higher sensitivity. LP-19, a short peptide of our previous design, had an “M-T” hook with obviously effective inhibitory activity against T20-resistant strains.

### 2.9. Structural Analysis of the Resistant Sites

There were more overlapping resistance sites for T20 and LP-40, with the important resistance sites including L33S, I37T, V38A, and N42T. We used the structures of T20/N39 (PDB: 5ZCX) [20] and LP-40/N44 (PDB: 5Y14) [15] to analyze the resistance mechanisms of these mutation sites. As shown in Figure 7A, Leu-33, Leu-34, and Val-38 formed a hydrophobic pocket, which had hydrophobic interactions with Leu-149. As for L33S and V38T, Ser-33 and Thr-38 broke this hydrophobic pocket due to the properties of serine, and due to threonine being polar. In addition, both valine and alanine are non-polar amino acids, while the substitution of Val-38 by a smaller-residue alanine reduced the hydrophobic contacts. As shown in Figure 7B, Ile-37 is located in the center of the three-stranded NHR helix, forming a “hydrophobic core”. Replacing Ile with Thr destroys this hydrophobic core, destabilizing the NHR trimer and thus affecting its binding to T20. In the crystal structure of T20/N39, Asn-42 can form a hydrogen bond with Gln-142, while, in the crystal structure of LP40/N44, Asn-42 forms a hydrogen bond with Glu-146 (the grey part in Figure 7C). Thr-42 destroys the hydrogen bond with Gln-142 and Glu-146. Asn-43 can form hydrogen bonds with Glu-137 and Gln-141, while Ser-43 can only form a hydrogen bond with Ser-138.

NDQEEDYN mutations occurred frequently in the CHR of Env of T20- and LP40-resistant mutants. We analyzed the effect of NDQEEDYN mutations on the stability of viral 6-HB based on the crystal structure of the HIVgp41 core structure (PDB number:1AIK) [5]. As shown in Figure 7D, aspartic acid was substituted by glutamic acid at the 121 site, and both of them could form a salt bridge with Lys-63. Although aspartic acid and glutamic acid are negatively charged residues, glutamic acid is larger than aspartic acid, so the salt bridge becomes shorter and the interaction between Glu-121 and Lys-63 become stronger. Asn-125 has no interaction with any amino acid, while Glu-125 can form a salt bridge with Lys-63. To verify the above analysis, we analyzed the interactions in the crystal structure of SFT/N36 (PDB number:3VIE) [21], in which the corresponding sites, 121 and 125, in the sequence of SFT, were Glu-121 and Glu-125. As shown in the gray part of Figure 7D, the conformations of Glu-121 and Glu-125 in SFT/N36 are different from that of C34/N36. Glu-121 can not only form more electrostatic attractions and salt bridges with Lys-63 but it can also form a hydrogen bond with Gln-66. Glu-125 can also form more electrostatic attractions with Lys-63, but the bond length is longer than C34/N36. In conclusion, D121E and N125E can increase the interaction between NHR and CHR, and this may be the reason for why the fusion and infectivity abilities of the NDQEEDYN mutant were increased. 

## 3. Discussion

We have previously demonstrated the importance of TRM in the antiviral activity of T20. The TRM region of the carboxyl-terminal group of T20 is rich in tryptophan sequences (WASLWNWF) and is highly hydrophobic. An excision of the TRM sequence causes an almost complete loss of T20 antiviral activity, and the T20-based lipopeptide (LP-40) was designed by replacing TRM with C16 fatty acids (Figure 1). The C16 fatty acid modification greatly enhances the stability of LP-40 bound to N39 to form the 6-HB structure. LP-40 significantly improved the binding affinity and the inhibitory activity compared to T20. LP-40 shows good prospects for clinical application. Considering the drug resistance problem of T20 in clinical applications, several questions remain: can LP-40 overcome the resistance weakness of T20? Is there potential for its further development? Therefore, in this study, we chose to induce HIV mutants under parallel conditions with T20 and LP-40 inhibitors and compared the similarities and differences between them via mutational analysis. We found that: 1. during the first month, LP-40 has a significantly higher resistance barrier for resistance development than T20; 2. as an exception to the well-recognized amino acid mutations at gp41 NHR 36-45 that reduce the inhibitory activity of CHR-derived peptides, mutations in L33S show stronger drug resistance; 3. the eight amino acid combination mutation (NDQEEDYN) in the CHR region is a compensatory mutation; and 4. LP-40 and T20 have similar mutation sites, so it is speculated that the same resistance might occur if LP-40 is used in a clinical setting.

We selected the IC_90_ concentration of LP-40 and T20 as the starting induction concentration. The rate of concentration increase was considered using the conditions of P24 quantification and cytopathic C8166 cells. Impressively, during the first month, peptide sequence differences led to the display of different resistance patterns before the screening. The proportion of the C8166 cytopathic effect under LP-40 drugs was much lower than that of T20, and the LP-40 uplift rate was forced to slow down. A comparison of the resistance curves shows that LP-40 has difficulty inducing resistance during in vitro selection, which means that it has a significantly higher barrier to resistance than T20. However, the resistance multiplicity of LP-40 increases similarly to T20 in the later stages, with no significant resistance advantage. Both T20 and LP-40 were selected in vitro for nearly 1 year, with combinatorial mutations in the gp41-encoding sequence. Phenotypic and replication kinetic analysis revealed that, in the case of both inhibitors, changes in the gp41 NHR sequence were the main mutations that reduced resistance to the inhibitors, while changes in gp41 CHR were the compensatory secondary mutations. The accumulation of multiple mutations in the gp41 NHR sequence eventually caused high levels of resistance to T20 and LP-40, and even single amino acid (L33S or I37T) substitutions significantly affected the sensitivity of both inhibitors.

It is reported that, during T-20 treatment, HIV-1 obtained the T-20 resistance mutation, especially as the interactions of the amino acid of gp41 NHR 36-45 (DIVQQQNNLL) and CHR are the main factors leading to resistance. In our study, in addition to sites 36–45, site 33 was also found. L33S appeared early in the T20 screening but later in the LP-40 resistance sites. From the structural analysis, under the L33S mutation, the non-polar amino acid Leu became the polar amino acids Ser and Thr, breaking the hydrophobic reaction with Leu-149, so it is speculated that the stability of the six-helix bundle is weakened. With the single-point mutation L33S, there are no evident changes in the entry capacity of the mutant L33S strain, but its fusion capacity is significantly reduced, so we can further verify that the mutant L33S affects its fusion process. A single point, L33S, reduced the sensitivity of T20 and LP-40 by 51- and 67-fold, respectively, indicating its important role. We found that a single-point mutation, I37T, reduced the sensitivity of T20 and LP-40 by 10- and 19-fold, respectively, and that this I37T also appeared in patients during T20 treatment [22,23,24,25], indicating the importance of this mutation point. Although I37T was only found in LP-40 mutants when we induced drug resistance, we analyzed that L33S appeared earlier in the T20 drug resistance process and that its drug resistance was much greater than that of I37T. However, I37T emerged early in LP-40-induced resistance, and it gradually replaced I37T with the more resistant mutation L33S as the drug concentration increased.

Peptides derived from the viral CHR sequence can bind to the viral NHR region to form a six-helix bundle, thus effectively preventing the occurrence of the viral membrane fusion process. In a resistance screen of the membrane fusion inhibitors T20 and LP-40, we found eight amino acid combination mutations (NDQEEDYN) located in the CHR region of gp41, which frequently accompanied the generation of NHR primary resistance mutation sites. This was also observed during in vitro screening with other peptides with a CHR sequence (data not reported). These eight amino acid combination mutations also appeared during the parallel in vitro selection of T20 and LP-40, so we infer that the mutations in the CHR region were not accidental but compensatory mutations. We constructed the strains containing these eight-point mutations and found that their fusion abilities were significantly enhanced compared with the wild-type strains. From their structures, both the D121E and N125E mutations can strengthen the combination of NHR, indicating that, under drug pressure, not only can mutations be observed in the NHR region but important compensatory mutations can also be found in the CHR region. We found an interesting phenomenon in that all the inhibitors except T20 and T1249 showed reduced sensitivities to the combinatorial mutation NDQEEDYN. Analyzing the inhibitor sequences, we found only the sequences of T20 and T1249 contained in the TRM of the tryptophan-enriched region, ^155^WASLWNWF^162^. We speculate that the antiviral activity was enhanced because the presence of the TRM region sequence in T20 and T1249 overcame weak binding due to eight amino acid mutations in the CHR of the virus. The lipopeptide LP-40 is a TRM sequence, with fatty acid (C16) replacing the carboxyl end of T20. Although the antiviral activity is enhanced, it has no advantage over the compensatory mutation resistance points. LP-19 binds to the pocket-2 region, the sequence of which is distant from the TRM region and not significantly affected by NDQEEDYN compensatory mutations, while the peptides with a sequence origin that is closer to the TRM region show a greater effect.

It is very important to conduct in vitro resistance screening before making clinical applications. First, this can predict the resistance barrier of the inhibitors; additionally, it can analyze possible resistance points and comprehensively analyze the development prospects of the inhibitors. Although the resistance barrier of LP-40 was significantly higher than the parent compound, T-20, the mutations induced by T20 and LP-40 largely overlapped, and most of the mutations showed significant cross-resistance. This shows that replacing the TRM sequence at the carboxy terminus of T20 by fatty acid (C16) significantly improves the inhibitory activity of HIV strains, but similar resistance may occur if it is used in a clinical setting. Such research not only enriches our understanding of the structure and function of the HIV-1 envelope protein but also deepens our understanding of the resistance mechanisms of membrane fusion inhibitors. This will help researchers to review the interactions between viruses and inhibitors from multiple points of view, and it provides new ideas for the development, improvement, and application of peptide-based membrane fusion inhibitors.

## 4. Materials and Methods

### 4.1. Peptide Synthesis

The C peptides, including T20, LP-40 [15], C34, SFT, T1249, T2635, and LP-19 [26], and the N peptide, N36, were synthesized using a standard solid-phase 9-fluorenylmethoxy carbonyl (FMOC) method as described previously. The N terminus and C terminus of these peptides were acetylated and amidated, respectively. They were purified using reverse-phase high-performance liquid chromatography (HPLC) to a purity of >95%. All peptides were reviewed to be of the correct amino acid composition via mass spectrometry.

### 4.2. Selection of T20- or LP40-Resistant Viruses

The in vitro selection of HIV-1 resistance to the peptide inhibitors T20 or LP-40 was performed as described previously. Briefly, the infectious viruses of HIV-1 NL4-3 were generated via the transfection of 293T cells with an encoding plasmid. C8166 cells were seeded at 1 × 10^4^ in RPMI 1640 medium containing 10% fetal bovine serum (FBS) on 12-well plates. Virus infection of the cells was tested in the presence or absence of diluted T20 or LP-40. Cells were incubated at 37 °C with 5% CO_2_ until an extensive cytopathic effect was observed [27,28,29]. Culture supernatants were harvested and used for the next passage on fresh C8166 cells, with a 1.5- to 2-fold increase in peptide concentrations. Two replicate wells were made for each sample, and the cells and supernatants were collected periodically and stored at −80 °C.

### 4.3. Mutation Determination and Site-Directed Mutagenesis

Viral RNA was extracted from the supernatant, and, after reverse transcription, the cDNA was amplified using PCR. These amplified DNA were connected to the T-Easy vector and transformed to Trans 2-Blue Chemically Competent Cells. These sequences were determined via sequencing [29,30]. The expression plasmids, which contained different HIV-1NL4-3 Env mutants, were constructed using DNA assembly (NEBuilder^®^ HiFi DNA Assembly Master Mix). In brief, two primers were used to amplify Env, and the other two primers were used to amplify the vector fragment. The products of PCR were purified with a 20–25 bp overlap in the terminal of Env and the vector fragment Env and the vector fragment were added to the DNA assembly master mix at 65 °C for 15 min and then transformed to Trans 2-Blue Chemically Competent Cells. Additionally, a single-point mutation was generated using double-stranded DNA templates and a selection of mutants with a restriction enzyme (DpnI) as described previously. The successful mutations were confirmed via sequencing.

### 4.4. Single-Cycle Infection Assay

The infectivity of HIV Env mutants and the inhibitory activities of membrane fusion inhibitors on mutants in TZM-bl cells were measured using a single-cycle infection assay as described previously [31]. Briefly, the HIV pseudovirus was packed using an Env-coding plasmid, and pSG3ΔEnv was co-transfected into HEK293T cells. pSG3ΔEnv encodes an Env-defective, luciferase-expressing HIV-1 genome. After the pseudoviruses were harvested and filtrated, the amount of pseudoviruses was fixed via P24 antigen, and their infectivity was determined in TZM-bl cells. In order to measure the inhibitory activities of the inhibitors, inhibitors diluted serially 3-fold were mixed with an equal volume of pseudoviruses, and then 100 µL 1 × 10^5^/mL TZM-bl cells were added to the mixture and incubated at 37 °C with 5% CO_2_ for 48 h. After that, TZM-bl cells were lysed using reporter lysis buffer, and IC_50_ values were calculated according to luciferase activity measured by luciferase assay reagents and a luminescence counter (Promega, Madison, WI, USA).

### 4.5. DSP-Based Cell–Cell Fusion Assay

The fusion activities of Env mutants were measured based on a dual-split-protein (DSP) system [32,33]. On the first day, 2 × 10^5^/mL HEK293T cells were seeded into a 96-well plate and 293FT cells were seeded into a 6 cm culture dish, and then these cells were incubated at 37 °C with 5% CO_2_. On the next day, the Env-coding plasmid and the DSP1-7 plasmid were co-transfected into HEK293T cells; on the third day, 3 × 10^5^/mL 293FT cells mixed with 17 ng/mL EnduRen live cell substrate (Promega), which expresses CXCR4/CCR5 and DSP8-11, were transferred to the HEK293T cells and the mixture of cells was spun down at 300× *g* to facilitate cell–cell contact. Luciferase activity was measured per hour until the sixth hour using luciferase assay reagents and a luminescence counter (Promega, Madison, WI, USA).

### 4.6. Western Blotting

The expression and processing profile of Env mutants in the cell and the incorporation and cleavage of Env mutants in pseudotype particles were examined using a Western blotting assay as described previously [32]. To acquire the cell lysates, on the first day, 4 × 10^5^/mL HEK293T cells were seeded in a 6-well plate; on the next day, the Env-coding plasmid was transfected into the HEK293T cells, and HEK293T cells were harvested after 48 h. These transfected cells were lysed for 30 min on ice and centrifuged at 20,000× *g* at 4 °C for 1 h to remove insoluble materials. The supernatants of the lysates were diluted and fixed to the same protein concentration using a BCA Protein Assay Kit (Thermo Scientific™, Rockford, IL, USA). Proteins in the lysates were separated using SDS-PAGE and transferred to a nitrocellulose membrane. After blocking with 5% nonfat dry milk solution in Tris-buffered saline (TBS, pH 7.4) at room temperature for 1 h, the washed membrane was incubated with a rabbit anti-gp120 polyclonal antibody (SinoBiological, Beijing, China) or the human anti-gp41 monoclonal antibody 10E8 overnight at 4 °C. The next day, the washed membrane was incubated with IRDye 680LT goat-anti-rabbit IgG or IRDye 800CW goat-anti-human IgG for 2 h at room temperature. β-actin was detected with a mouse anti-β-actin monoclonal antibody (Sigma, St. Louis, MO, USA) and IRDye 680LT donkey-anti-mouse IgG.

## Figures and Tables

**Figure 1 ijms-23-06638-f001:**
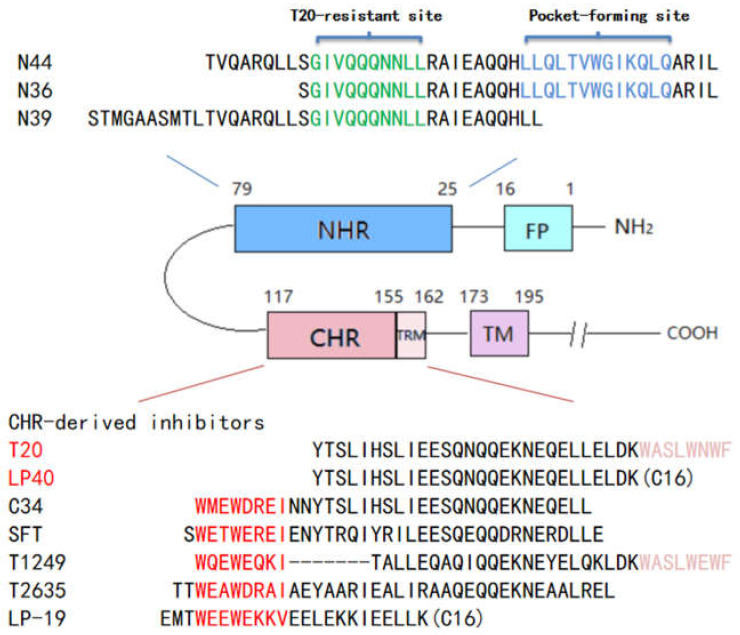
Schematic illustration of HIV gp41 and peptide fusion inhibitor. FP, fusion peptide; NHR, N-terminal heptad repeat; CHR, C-terminal heptad repeat; TRM, tryptophan-rich motif in the CHR; TM, transmembrane domain; CT, cytoplasmic tail. The sequences corresponding to the T20-resistant site and the pocket-forming site are marked in green and blue, respectively, on the NHR-derived peptide N36. The sequences corresponding to the pocket-binding site and TRM are marked in red and purple, respectively, on the CHR-derived peptide C34.

**Figure 2 ijms-23-06638-f002:**
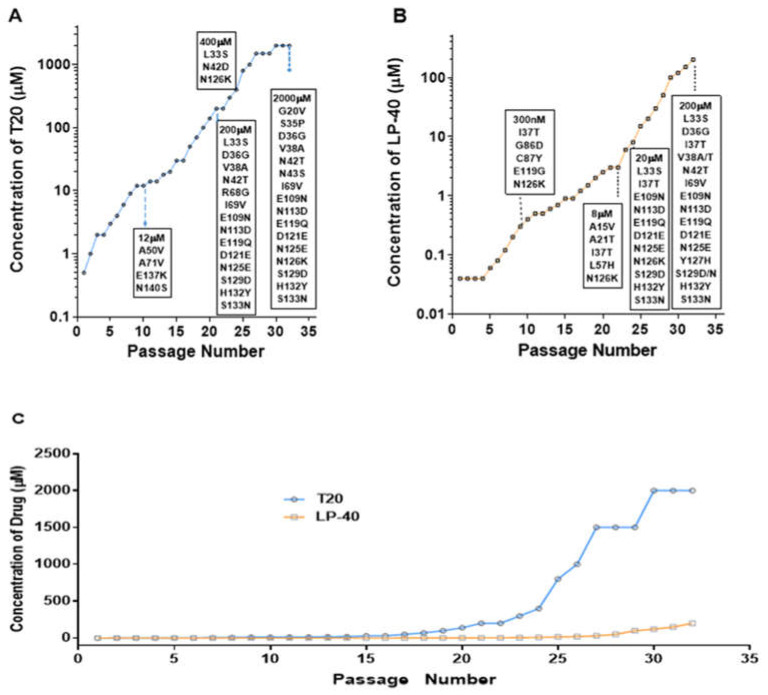
Introduction of T20- and LP40-resistant HIV-1. (**A**) The dose-escalating selections were conducted for a total of 32 passages, with T20 concentrations ranging from 500 nM to 2000 µM, and (**B**) for 32 passages, with LP40 concentrations ranging from 40 nM to 200 µM. At the indicated passage number, viral RNA was extracted from the supernatant, and gp41- and gp120-coding regions were sequenced. The mutations of gp41 are listed in the black frames. (**C**) HIV-1 NL4-3 was passaged in the presence of an increasing concentration of the inhibitors in C8166 cells.

**Figure 3 ijms-23-06638-f003:**
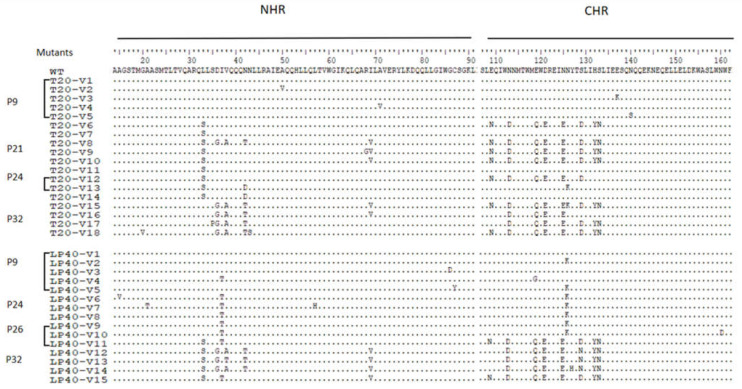
T20- and LP40-induced mutations in the gp41 region of HIV-1NL4-3. The amino acid sequences of wild-type (WT) and selected mutants are aligned. The positions of selected substitutions are in bold, and the numbering is according to that of HIV-1HXB2 gp41. WT, wild-type.

**Figure 4 ijms-23-06638-f004:**
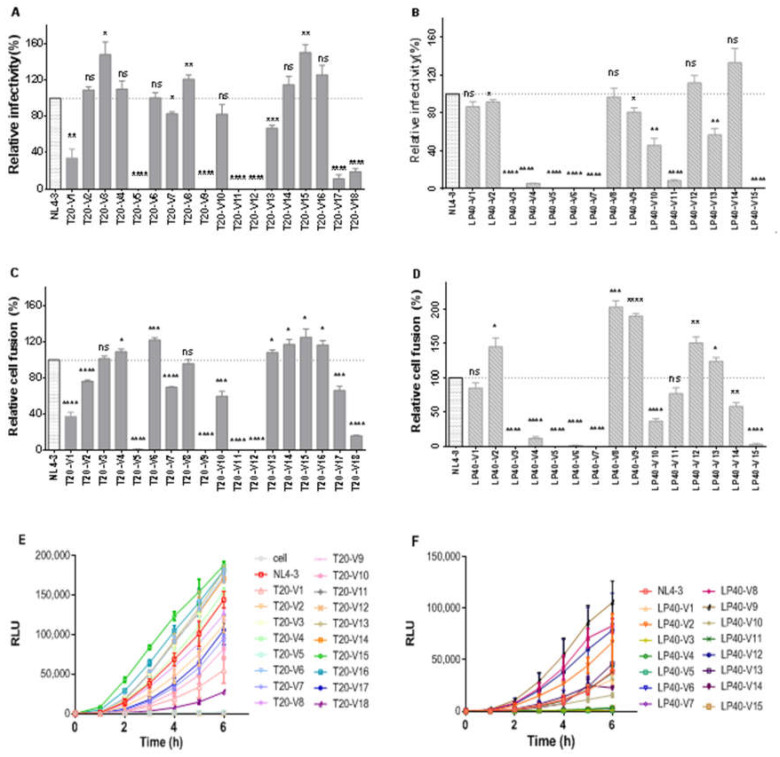
The functional Env profiles of T20- and LP40-resistant mutants. (**A**,**B**) Relative infectivity of HIV-1NL4-3 and T20- or LP40-resistant mutants, as determined via a single-cycle infection assay. These pseudoviruses were normalized to a fixed amount against P24 antigen, and viral infectivity was tested in TZM-bl cells. The luciferase activity was measured and corrected for background. (**C**,**D**) HIV-1 mutants’ Env-mediated cell–cell fusion measured using a dual split-protein assay (DSP). For both entry and fusion data, the luciferase activity of wild-type HIV-1NL4-3 (WT) was treated as 100%, and the relative activities of other mutant viruses were calculated accordingly. The experiments of infectivity and fusion were repeated three times, and these data are expressed as means with SD. Statistical comparison was conducted using a *t*-test, *, *p* < 0.05; **, *p* < 0.01; ***, *p* < 0.001; ****, *p* < 0.0001; ns, not significant. (**E**,**F**) Kinetics of HIV-1 Env-mediated cell–cell fusion, determined using a DSP-based assay.

**Figure 5 ijms-23-06638-f005:**
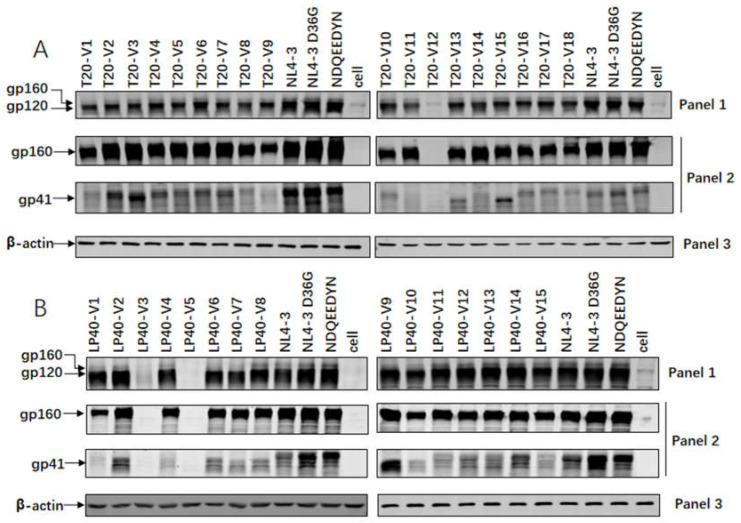
Expression and processing profiles of HIV-1 NL4-3 Env, detected using Western blotting. The (**A**) T20- and (**B**) LP40-resistant mutant glycoproteins in the lysates of transfected cells were detected with a rabbit anti-gp120 polyclonal antibody (panel 1) and 10E8 (panel 2). The β-actin protein was detected as an internal control (panel 3). Env incorporation in (**C**) T20- and (**D**) LP40-resistant pseudoviruses was determined using immunoblotting. Concentrated virions were re-suspended in RIPA lysis buffer, then normalized to a fixed amount using P24 antigen and subjected to SDS-PAGE. A rabbit anti-gp120 polyclonal antibody (panel 4) and 10E8 (panel 5) were used to probe gp160, gp120, and gp41. As an internal control, the P24 antigen was detected with a rabbit anti-HIV P24 polyclonal antibody (panel 6). The experiments were repeated three times, and representative data are shown.

**Figure 6 ijms-23-06638-f006:**
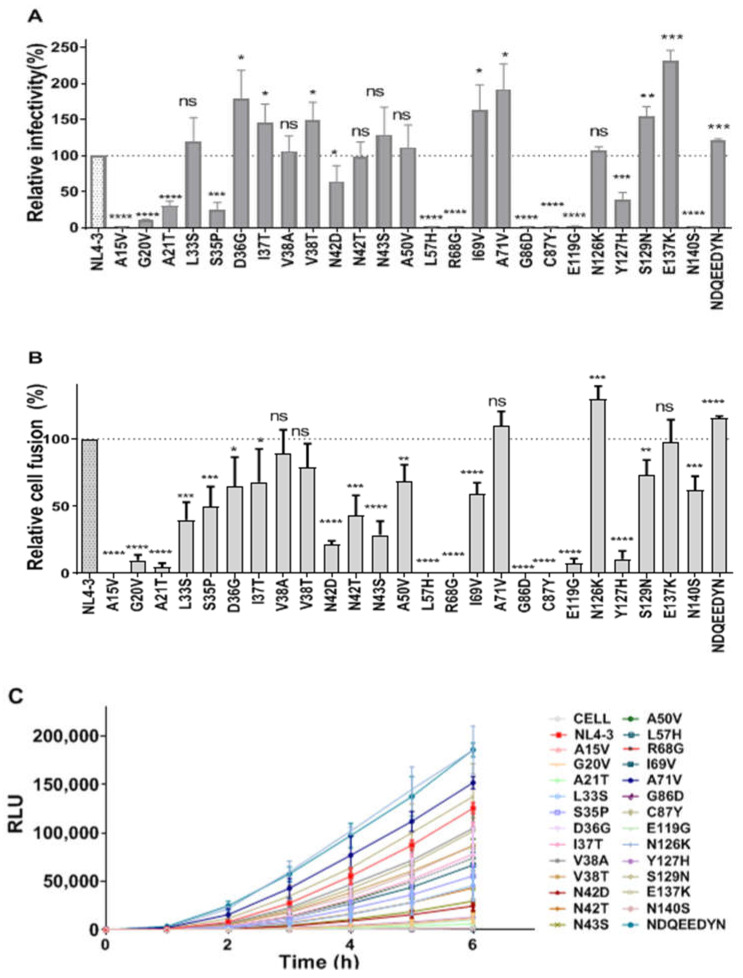
Effects of T20- and LP40-induced mutations on the functionality of HIV-1 Env. (**A**) Relative infectivity of HIV-1NL4-3 and its mutants, determined using a single-cycle infection assay. (**B**) HIV-1 Env-mediated cell–cell fusion, determined using a dual split-protein assay (DSP). For both entry and fusion data, the luciferase activity of wild-type HIV-1NL4-3 (WT) was treated as 100%, and the relative activities of other mutant viruses were calculated accordingly. The experiments of infectivity and fusion were repeated three times, and these data are expressed as means with SD. A *t*-test was performed to judge the significance of the difference between the WT and the mutants. *, *p* < 0.05; **, *p* < 0.01; ***, *p* < 0.001; ****, *p* < 0.0001; ns, not significant. (**C**) Kinetics of the HIV-1 Env-mediated cell–cell fusion, determined using a DSP-based assay.

**Figure 7 ijms-23-06638-f007:**
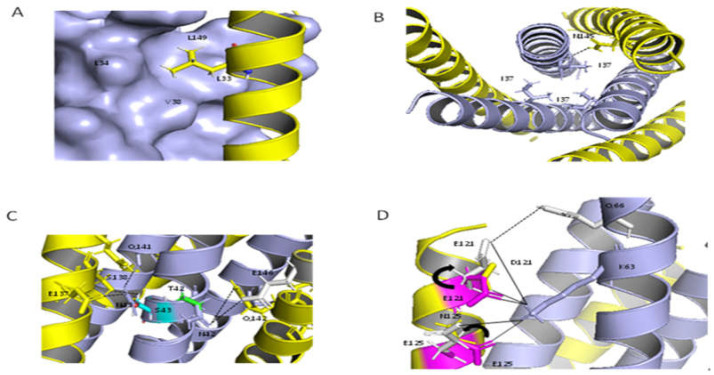
Structural basis of HIV-1 resistance to T20 and LP40. The effects of the resistance mutations on the binding stabilities of T20 or LP40 were analyzed using the crystal structure of T20/N39, shown as a ribbon model. Three N39 peptides are colored in blue, and three T20 peptides are colored in yellow. (**A**,**B**) The residues Leu33, Leu34, and Val38 are shown on the surface; Ile37 (colored in blue) and their interacting residues on N39, Leu149, and Asn147 (colored in yellow) are shown as stick models with labels. (**C**) Asn42 and Asn43 (colored blue), and their mutations, Thr42 and Ser43 (colored green and cyan), Glu137, Ser138, Glu142, and Gln144 on N39 (colored in yellow), and Glu146 on N44 (colored in gray), are shown as stick models with labels. (**D**) The effects of NDQEEDYN on the interactions of the viral NHR and CHR helices were analyzed using the crystal structure of N36/C34, shown as a ribbon model. N36 peptides are colored in blue, and the C34 peptides are colored in yellow. Asp121 (colored in yellow) and its mutation Glu121 (colored in magenta), and Asn125 (colored in yellow) and its mutation Glu125 (colored in magenta) are shown as stick models with labels. Their interacting residues on C34 and Lys63 (colored blue) are shown as stick models with labels. In order to confirm the interaction between Glu121, Glu125, and Lys63, we analyzed these interactions in SFT/N36; the conformations of Glu121 and Glu125 (colored in gray) in SFT are different from Glu121 and Glu125 (colored in magenta) in C34. The solid black lines indicate salt bridges, and the dashed lines indicate hydrogen bonds.

**Table 1 ijms-23-06638-t001:** Resistance profiles of T20- and LP40-resistant mutants to T20 and LP-40. The experiment was performed in triplicate and repeated three times. The background colour to distinguish between different concentrations.

	T20		LP-40	
Pseudovirus	IC_50_ (nM)	Fold	IC_50_ (nM)	Fold
NL4-3	55.01 ± 32.65	1.00	29.68 ± 4.52	1.00
T20-V1	57.69 ± 7.23	1.05	40.07 ± 8.62	1.35
T20-V2	146.70 ± 39.40	2.67	59.24 ± 15.90	2.00
T20-V3	78.92 ± 12.10	1.43	70.51 ± 8.56	2.38
T20-V4	53.13 ± 14.90	0.97	76.63 ± 19.66	2.58
T20-V6	3853.00 ± 516.06	70.04	3513.22 ± 631.31	118.37
T20-V7	2808.17 ± 457.53	51.05	2725.11 ± 211.71	91.82
T20-V8	5275.34 ± 1100.34	95.90	>6250	>227.43
T20-V10	3335.50 ± 432.90	60.63	6080.78 ± 1172.18	204.88
T20-V13	4010.67 ± 570.83	72.91	3185.11 ± 206.85	107.32
T20-V14	4429.42 ± 966.54	80.52	2895.78 ± 85.08	97.57
T20-V15	2089.66 ± 247.94	37.99	>6250	>227.43
T20-V16	1741.55 ± 229.44	31.66	>6250	>227.43
T20-V17	2891.28 ± 881.88	52.56	5167.89 ± 1020.06	174.12
T20-V18	6146.11 ± 475.36	111.73	>6250	>227.43
LP40-V1	59.76 ± 16.33	1.09	44.85 ± 4.39	1.51
LP40-V2	71.37 ± 4.32	1.30	373.61 ± 29.84	12.59
LP40-V8	1151.16 ± 137.30	20.93	2883.89 ± 1070.02	97.17
LP40-V9	1270.70 ± 314.74	23.10	2885.45 ± 430.88	97.22
LP40-V10	1261.73 ± 264.08	22.94	2746.00 ± 634.40	92.52
LP40-V11	4348.78 ± 521.53	79.05	3333.78 ± 977.77	112.32
LP40-V12	5868.67 ± 978.46	106.68	>6250	>227.43
LP40-V13	5167.09 ± 2461.48	93.93	>6250	>227.43
LP40-V14	4911.25 ± 1996.92	89.28	>6250	>227.43

Data are expressed as means ± SD. Fold-change in IC_50_ was determined relative to the WT (HIV-1_NL4-3_) level.

**Table 2 ijms-23-06638-t002:** Resistance profiles of HIV-1 mutants to T20 and LP-40. The experiment was performed in triplicate and repeated three times. Data are expressed as means ± SD. Fold change in the IC_50_ was determined relative to the WT (HIV-1_NL4-3_) level.

	T20		LP-40	
Pseudovirus	IC_50_ (nM)	Fold	IC_50_ (nM)	Fold
NL4-3	67.34 ± 15.33	1.00	31.54 ± 10.42	1.00
G20V	49.12 ± 10.94	0.73	204.78 ± 47.22	6.49
A21T	58.85 ± 9.24	0.87	26.75 ± 8.36	0.85
L33S	3448.00 ± 564.06	51.20	2115.78 ± 533.73	67.08
S35P	137.56 ± 10.60	2.04	21.63 ± 5.15	0.69
D36G	6.00 ± 1.68	0.09	2.22 ± 0.57	0.07
I37T	710.62 ± 156.89	10.55	514.91 ± 74.31	16.33
V38A	1544.24 ± 21.96	22.93	3487.11 ± 423.38	110.56
V38T	991.11 ± 215.54	14.72	1451.90 ± 236.99	46.03
N42D	164.10 ± 26.82	2.44	109.08 ± 18.40	3.46
N42T	413.44 ± 81.98	6.14	1267.43 ± 167.03	40.18
N43S	540.06 ± 91.86	8.02	356.87 ± 117.38	11.31
A50V	162.15 ± 69.39	2.41	58.83 ± 13.68	1.87
I69V	71.39 ± 23.19	1.06	29.40 ± 4.59	0.93
A71V	46.50 ± 12.52	0.69	43.25 ± 8.19	1.37
N126K	116.81 ± 37.35	1.73	304.39 ± 50.53	9.65
Y127H	23.71 ± 2.63	0.35	5.74 ± 1.86	0.18
S129N	70.32 ± 19.83	1.04	34.03 ± 10.04	1.08
E137K	68.50 ± 20.14	1.02	42.64 ± 11.73	1.35

**Table 3 ijms-23-06638-t003:** Resistance profiles of mutations that belong to LP40-V12 to diverse HIV-1 peptide fusion inhibitors. The experiment was performed in triplicate and repeated three times. Data are expressed as means. Fold change in the IC_50_ (nM)was determined relative to the WT (HIV-1_NL4-3_) level.

	T20		LP40		C34		SFT		T1249		T2635		LP-19	
Pseudovirus	IC_50_	Fold	IC_50_	Fold	IC_50_	Fold	IC_50_	Fold	IC_50_	Fold	IC_50_	Fold	IC_50_	Fold
NL4-3	71.88	1.00	27.32	1.00	0.80	1.00	1.56	1.00	1.14	1.00	0.73	1.00	0.0800	1.00
I37T	710.62	9.89	514.91	18.85	14.33	17.84	11.25	7.21	1.38	1.21	0.97	1.33	0.1298	1.62
L33S	3448.00	47.97	2115.78	77.43	5.55	6.90	4.67	3.00	13.01	11.41	1.43	1.97	0.1289	1.61
I69V	71.39	0.99	29.40	1.08	1.38	1.72	1.67	1.07	0.93	0.82	0.75	1.04	0.0800	1.00
L33S/I37T	5847.22	81.35	2979.00	109.04	110.54	137.60	22.97	14.72	81.48	73.47	1.51	2.07	0.1000	1.25
V38A/N42T	3348.55	46.59	4074.78	149.15	109.91	137.39	116.19	74.48	27.10	23.77	0.56	0.77	0.0530	0.66
L33S/V38A/N42T	4915.34	68.38	4990.44	182.67	912.42	1140.53	260.02	166.68	258.83	227.04	1.30	1.78	0.0444	0.56
NDQEEDYN	62.45	0.87	100.20	3.67	6.68	8.32	8.03	5.15	1.07	0.94	1.94	2.67	0.1345	1.68
NDQEEDYN/L33S/I37T	3794.89	52.80	3278.89	120.00	679.76	846.17	110.30	70.71	165.92	145.54	5.44	7.49	0.2433	3.04
NDQEEDYN/L33S/V38A/N42T	4781.45	66.52	>6250	>228.77	2497.78	3122.23	738.73	473.54	343.28	301.12	5.73	7.85	0.1180	1.48

## Data Availability

Not applicable.

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
