# Peer review of "In Vitro Selection and Characterization of HIV-1 Variants with Increased Resistance to LP-40, Enfuvirtide-Based Lipopeptide Inhibitor"

_ijms, 2022, doi:10.3390/ijms23126638_

Round 1
Reviewer 1 Report
Membrane-anchored synthetic or genetically-encoded C-peptides are more potent inhibitors of viral fusion than soluble variants. Previously, this group synthetized LP-40, demonstrated its higher inhibitory activity over T-20, and characterized some resistance mutations in HIV gp41. In this study, He with coauthors has characterized mutations in gp41 that arouse from dose-escalating treatment of HIV with T-20 and its lipophilic derivative LP-40. This is large and comprehensive study covering sequence analysis of generated mutants, characterization of infectivity, fusogenicity, expression, cross-resistance of individual or combined mutations in gp41. As a result, new insights in mechanism of T-20-derived peptides are presented, such as resistance mutation L33S and compensatory 8 ac changes NDQEEDYN in CHR. The collected data are informative in terms of predicting peptide resistant during HIV treatment, add some novelties, and paper can be recommended for publication after certain clarifications and minor improvements.
1. The levels of Env expression are presented only for cell lysates (Fig.5), though in Method section there is an indication that it’s done for viral particles as well (line 564-565). There are some discrepancies between infectivity and fusion data for some mutants, for example N140S. Did you measure incorporation of mutant Envs into pseudoviral particles? One of the explanations for this difference could be low or enhanced Env pseudotyping. This aspect should be clarified and at least discussed in the last section.
2. It is not clear how almost dead Envs were selected, for example, T20-V5 or LP40-V3 and others? (Figure 4)
3. Many figures of low quality and the letters standing for amino acids are barely readable.
4. No reference to Figure 4 E-F
5. Lines 33-42, the sentences were doubled
6. Please, check typos and sloppy phrases throughout the text in lines 51, 59, 71, 73, 94 (absent abbreviation), 105, 214, 218, 254
7. TRM in the abstract at the first appearance is better to give in full name.
Reviewer 2 Report
The present manuscript compares resistance induction by the classical Enfuvirtide (T-20) HIV fusion inhibitor and by LP-40,Enfuvirtide -Based Lipopeptide Inhibitor. The resistance profile of the two related compounds is overall very similar, with extensive cross-reactivity and a similar pattern of functional consequences of the resistance-associated mutations (RAM).
The study is well done, although it is a pity that apparently only one attempt was done. Therefore it is difficult to assess whether subtle differences in RAM between the two compounds are just the consequence of chance events. The results would have been more convincing if the resistance profile in several duplicate cultures of the same HIV strain was assessed and if a second strain of a different clade had been included in parallel.
The text is sometimes difficult to read and requires thorough revision of the English language and writing style.
The authors claim a higher barrier to resistance for LP-40, as compared to T-20, but that is very questionable, since the fold increase in IC50 for both compounds is ultimately very similar (taking into account the lower baseline IC50 of LP-40).
In view of all these observations, it is highly uncertain that LP-40 could have a future in the clinic, since, in addition, there are many highly active combination that can be administered orally. But that doesn’t diminish the scientific value of the present manuscript.
Round 2
Reviewer 2 Report
The authors have sufficiently addressed my remarks.